# Inorganic Polyphosphate: An Emerging Regulator of Neuronal Bioenergetics and Its Implications in Neuroprotection

**DOI:** 10.3390/biom15081060

**Published:** 2025-07-22

**Authors:** Marcela Montilla, Norma Pavas-Escobar, Iveth Melissa Guatibonza-Arévalo, Alejandro Múnera, Renshen Eduardo Rivera-Melo, Felix A. Ruiz

**Affiliations:** 1Medical School, Universidad Cooperativa de Colombia, Villavicencio 500003, Colombia; norma.pavas@campusucc.edu.co (N.P.-E.); renshen.riveramel@campusucc.edu.co (R.E.R.-M.); 2Behavioral Neurophysiology Laboratory, Physiological Sciences Department, School of Medicine, Universidad Nacional de Colombia, Bogotá 110110, Colombia; imguatibonzaa@unal.edu.co (I.M.G.-A.); famunerag@unal.edu.co (A.M.); 3Instituto de Investigación e Innovación Biomédica de Cádiz (INiBICA) and “Department of Biomedicine, Biotechnology, and Public Health, Medical School”, Universidad de Cádiz, 11003 Cadiz, Spain

**Keywords:** inorganic polyphosphate (polyP), mitochondrial bioenergetics, neuroprotection, neurodegenerative diseases, therapeutic target

## Abstract

Inorganic polyphosphate (polyP) is an evolutionarily conserved polymer that has recently gained relevance in neuronal physiology and pathophysiology. Although its roles, such as mitochondrial bioenergetics, calcium homeostasis, and the oxidative stress response, for example, are increasingly recognized, its specific implications in neurological disorders remain underexplored. This review focuses on synthesizing the current knowledge of polyP in the context of central nervous system (CNS) diseases, highlighting how its involvement in key mitochondrial processes may influence neuronal survival and function. In particular, we examine recent evidence linking polyP to mechanisms relevant to neurodegeneration, such as the modulation of the mitochondrial permeability transition pore (mPTP), regulation of amyloid fibril formation, and oxidative stress responses. In addition, we analyze the emerging roles of polyP in inflammation and related cell signaling in CNS disorders. By organizing the existing data around the potential pathological and protective roles of polyP in the CNS, this review identifies it as a candidate of interest in the context of neurodegenerative disease mechanisms. We aim to clarify its relevance and stimulate future research on its molecular mechanisms and translational potential.

## 1. Introduction

Inorganic polyphosphate, or polyP, is an intriguing linear polymer composed of orthophosphates. It has existed since the dawn of evolution and is found in both prokaryotic and eukaryotic organisms. PolyP plays a crucial role in maintaining the cellular balance, managing energy metabolism, and responding to stress [1,2,3]. While we are beginning to understand its importance in various biological systems, its specific functions in the central nervous system (CNS) remain incompletely understood. This raises intriguing questions about how polyP contributes to neuronal energetic processes and protective mechanisms [4]. In this article, we will delve into the latest findings on the role of polyP in neuronal physiology, explore its connection to mitochondrial function, and discuss its implications for neurodegenerative diseases.

The CNS is highly dependent on energy metabolism, which takes place mainly in the mitochondria, and is essential for keeping the redox state under control [4]. Recent studies suggest that polyP plays a multifaceted role in mitochondrial function, including the regulation of mitochondrial permeability, modulation of proteins involved in oxidative metabolism, and interactions with proteins governing mitochondrial biophysics. Their presence and chain length determine mitochondrial health and the cell’s ability to respond to stress, making them a potential target for therapies for diseases involving mitochondrial dysfunction [4,5,6,7]. Through these mechanisms, polyP may contribute to the regulation of neuronal bioenergetics and enhance cellular resilience to stress [7].

In this context, polyP is linked to several diseases, including Alzheimer’s disease, Parkinson’s disease [8], and amyotrophic lateral sclerosis (ALS) [9,10]. This connection indicates that polyP plays a role in regulating calcium levels, promoting the aggregation of harmful proteins, and contributing to neuroinflammation. The protective effects of polyP likely stem from its interactions with cell signaling pathways that are crucial for cell survival, making it a promising target for therapies aimed at preventing or treating CNS disorders.

This review aims to consolidate current knowledge about its function in neuronal energy processes and neuroprotection, emphasizing its growing significance in CNS-related diseases and its potential influence on biomedical research and the development of new treatments.

## 2. Structural Characteristics of PolyP

PolyP is a linear, anionic polymer composed of multiple orthophosphate (Pi) units linked by high-energy phosphoanhydride bonds (Figure 1). The length of polyP chains varies significantly, ranging from as few as three phosphate residues to several hundred, depending on the biological or synthetic context [1,2]. This polymer is recognized as a multifunctional structure capable of acting as a polyanionic protein scaffold that contributes to a wide array of cellular processes [2].

The polymerization of polyP, driven by the formation of phosphoanhydride bonds, results in diverse structural configurations, including linear, cyclic, branched, and even supramolecular assemblies. In synthetic forms, polyP chains can extend to several thousand orthophosphate units [12]. Considering that a phosphate monomer has an approximate length of ~2.7 Å, it is estimated that linear polyP chains in bacteria and mammals can reach lengths up to 300 nm [13,14].

In biological systems, polyP is rarely found in its free form. Instead, it typically associates with divalent cations and proteins, a property that facilitates its condensation within specialized subcellular compartments such as acidocalcisomes in protozoa and dense granules in mammalian platelets. Within these organelles, polyP plays key roles in regulatory and storage functions [1].

PolyP is highly conserved across all domains of life—from bacteria to higher eukaryotes—underscoring its evolutionary and functional significance [1,2]. Studies using animal models, particularly rats and mice, have demonstrated notably elevated concentrations of polyP in the heart and brain, with levels reaching up to threefold higher than those observed in the liver [1,15]. However, despite its widespread distribution, substantial knowledge gaps remain concerning its metabolism, regulatory mechanisms, intracellular concentration, chain length, and precise localization within mammalian cells.

A key functional group of polyP is its high affinity for calcium ions (Ca^2+^), with a dissociation constant (Kd) in the low nanomolar range. This high affinity implies that under physiological conditions, polyP exists predominantly in a calcium-saturated form. Given that Ca^2+^ concentrations in the extracellular (2.2–2.6 mM) and intracellular (0.1–1.0 mM) spaces significantly exceed its Kd, this interaction critically influences the biophysical properties of polyP, promoting the formation of insoluble aggregates in physiological spaces. Such structural transitions impact both the functional roles and the bioavailability of polyP across various cellular compartments [16,17].

## 3. Metabolism and General Functions of PolyP

### 3.1. PolyP in Bacteria

In bacteria, the synthesis and catabolism of inorganic polyP are tightly regulated by specific enzymatic systems, with two main types of polyphosphate kinases (PPKs, Figure 2) playing a central role: PPK1 and PPK2. The PPK1 enzyme catalyzes the transfer of terminal phosphate groups from ATP molecules to an elongating polyP chain, thereby promoting polymer extension. In contrast, PPK2 utilizes polyP as a phosphate donor to phosphorylate nucleoside diphosphates such as GDP and ADP, thus playing a key role in nucleotide regeneration under conditions of energy limitation [18,19].

PolyP functions as a multifunctional molecule within bacterial cells, contributing to a broad spectrum of adaptive and defensive processes. It has been shown to participate in ion homeostasis, metabolic regulation, and protection against environmental stresses, including thermal shock, ultraviolet radiation, antibiotic exposure, and oxidative damage. Furthermore, polyP has been implicated in the formation of biofilms and the expression of virulence factors in pathogenic species such as *Pseudomonas aeruginosa* and *Helicobacter pylori* [20,21].

The regulated degradation of polyP is mediated by exopolyphosphatases (PPXs, Figure 2), a group of enzymes responsible for the sequential hydrolysis of phosphate residues from the terminal end of the polymer. These enzymes facilitate the progressive release of inorganic phosphate (Pi) and are functionally classified into two main types—PPX1 and PPX2—based on differences in their catalytic domains and substrate specificities [2].

### 3.2. PolyP in Unicellular Eukaryotes

In these cells, the degradation of polyP is primarily mediated by the enzyme endopolyphosphatase 1 (PPN1, Figure 2), which exhibits both endopolyphosphatase and exopolyphosphatase activities. PPN1 is distributed across multiple subcellular compartments, including the cytosol, nuclear envelope, mitochondrial membranes, and vacuoles [22,23].

Both PPN1 and the exopolyphosphatase PPX1 require the presence of Mg^2+^ ions to achieve optimal catalytic activity, while their function is not enhanced by Ca^2+^ ions. The optimal pH for their enzymatic action is approximately 7.0, reflecting a specific biochemical adaptation for the efficient hydrolysis of polyP under neutral intracellular conditions [22].

Among the most significant organelles involved in polyP accumulation in unicellular eukaryotes are acidocalcisomes, first described in *Trypanosoma cruzi* [14,24]. These organelles, enriched in polyP and Ca^2+^, have also been documented in bacteria and human cells, suggesting a convergent evolutionary development of specialized compartments dedicated to the storage and regulation of phosphate and divalent cations [2]. Their involvement in key physiological functions, such as intracellular calcium signaling and osmotic balance regulation, underscores the role of polyP as more than a simple phosphate reservoir, positioning it as a multifunctional regulator of cellular homeostasis [2].

### 3.3. PolyP in Mammals

In contrast to unicellular organisms, in which the enzymatic pathways involved in the synthesis and degradation of polyP have been extensively characterized, the metabolism of polyP in mammals has, in recent years, attracted increasing interest from the scientific community. This polymer exhibits notable variability in terms of its concentration, subcellular distribution, and chain length [25].

PolyPs are particularly enriched in mitochondria, with chain lengths ranging from 50 to 800 residues. Their presence has been reported in a variety of tissues, including the brain, heart, kidney, liver, and lungs [1,15,26]. In rat hepatocytes, the highest polyP concentrations have been reported in the nucleus (89 ± 7 μM), followed by the plasma membrane (43 ± 3 μM), cytosol (12 ± 2 μM), and mitochondria (11 ± 0.6 μM), with the lowest levels observed in microsomes (4 ± 0.5 μM) [15,27]. Furthermore, polyP has been detected at high concentrations in immortalized cell lines such as NIH3T3 fibroblasts, Vero renal epithelial cells, and CD4^+^ T lymphocytes [15].

Functionally, polyP plays a key role in the regulation of mitochondrial bioenergetics. Its accumulation in mitochondria is associated with inorganic phosphate and ATP homeostasis, with both its concentration and chain length modulated by the F_0_F_1_-ATP synthase, an essential enzyme for oxidative phosphorylation [28,29]. Recent findings suggest that this enzyme is also involved in polyP synthesis, although the complete set of enzymes responsible for polyP metabolism in mammals remains unidentified [15,28,30,31]. Notably, the synthesis of polyP may occur directly from inorganic phosphate, potentially involving membrane components and bypassing ATP pools [15]. 

PolyP degradation, on the other hand, is mediated by alkaline phosphatases, membrane-bound glycoproteins with phosphoesterase activity that are widely expressed in mammalian tissues [32]. In addition, NUDT3 has been identified as the first human endopolyphosphatase [33]. It is a Zn^2+^-dependent enzyme that shows robust endopolyphosphatase activity also with Mn^2+^ [33].

A crucial enzyme involved in polyP metabolism is h-Prune, a mammalian short-chain phosphohydrolase associated with tumor progression and metastasis [28,34]. Although h-Prune efficiently degrades short polyP chains (up to four phosphate units), it is ineffective against medium and long polyP chains (13–160 units), suggesting the presence of alternative enzymatic mechanisms specialized in managing longer-chain polyPs [29,31]. Recent studies have demonstrated that recombinant h-Prune lacks exopolyphosphatase activity on polyP chains ranging from 13 to 160 phosphate units, which are prevalent in mammalian cells, indicating that other enzymes may be responsible for the metabolism of these longer polyP chains [28]. Furthermore, h-Prune’s role in polyP metabolism appears to be conserved across species, as evidenced by similar findings in Drosophila melanogaster [1].

Various specialized eukaryotic cells, such as mast cells, astrocytes, and thrombocytes, have evolved active secretion mechanisms for polyP from intracellular storage organelles [35]. This capacity for regulated exocytosis highlights the physiopathological relevance of polyP in diverse biological contexts.

In the hemostatic context, polyP released from activated platelets has been shown to activate coagulation factor XII and promote the formation of fibrin fibers at procoagulant sites [36,37]. Additionally, polyP participates in the induction of inflammatory responses upon its release from mast cells, thereby modulating innate immune functions [38]. In the autonomic nervous system, polyP has been proposed to act as a gliotransmitter, facilitating intercellular communication mediated by astrocytes [32]

PolyP induces apoptosis in human plasma and myeloma cells by inhibiting immunoglobulin secretion, triggering the externalization of phosphatidylserine, activating caspase 3, and causing cell cycle arrest. These effects are specific to plasma and B lymphoid cells, sparing normal B, T, and non-lymphoid cells [39].

The effects of polyPs on apoptosis are highly context- and cell type-dependent. For example, polyP promotes apoptosis in plasma cells but inhibits it in osteoblast-like cells, where it instead stimulates proliferation and migration [39,40].

Notably, recent findings indicate that polyP can bind to basic residue-rich sequences within the SARS-CoV-2 Spike protein, thereby inhibiting its interaction with the ACE2 receptor. This suggests a potential therapeutic application of polyP in preventing viral infections such as COVID-19 [41].

Concurrently, polyP has been implicated as a mediator of post-translational modifications, specifically through lysine polyphosphorylation. This mechanism is involved in critical cellular processes such as DNA supercoiling and ribosome biogenesis, highlighting its potential regulatory role in eukaryotic systems [42,43].

In summary, Figure 3 depicts the biological functions of polyP in prokaryotes and eukaryotes, highlighting both domain-specific roles and shared functions. In prokaryotes, polyP is involved in essential processes such as motility, cell survival, biofilm formation, stress responses, metal ion chelation, and intracellular pH regulation. In eukaryotes, polyP is associated with complex functions, including blood clotting, angiogenesis, amyloid fiber formation, neurotransmission and gliotransmission, osteoblast activity, ion channel regulation, DNA repair, and apoptosis. In both domains, polyP serves as a key component in energy metabolism, cell growth, development, and intracellular phosphate storage.

## 4. Multifunctional Nature of PolyP in Neurons

The recent literature has firmly established the multifunctional nature of polyP in neurons and neurodegeneration (Figure 4), characterizing it as a polyanionic protein scaffold actively involved in a wide array of cellular processes [2]. Among its most prominent roles is the regulation of mitochondrial permeability transition pore (mPTP) opening and intracellular calcium signaling, both of which are critical for mitochondrial metabolism and the prevention of Ca^2+^-dependent cell death [44]. Within the mitochondrial environment, polyP also contributes to the maintenance of the membrane potential, underscoring its essential role in cellular functionality [44].

On the other hand, extracellular polyP, released by astrocytes bearing ALS/FTD-associated mutations and by activated mast cells, acts as a central mediator of motoneuron degeneration. Astrocytes release long-chain polyP that is stored in lysosomal vesicles in response to intracellular calcium elevations mediated by P2Y1 purinergic receptors, leading to neuronal hyperexcitability and cell death. Mast cells, in turn, release medium-chain polyP along with inflammatory mediators that, although they exhibit limited direct neurotoxicity, can trigger the secondary release of neurotoxic polyP from astrocytes, thereby amplifying neuronal damage. This model highlights extracellular polyP of both glial and immune origin as a key factor in the pathological progression of ALS and FTD [10].

### 4.1. Protein Scaffolding Functions

The biological significance of polyP has been largely attributed to its capacity to interact with specific proteins, functioning as a polyanionic scaffold that modulates diverse cellular processes [2]. Its scaffolding functions are crucial for protein folding, protection against stress, the regulation of mitochondrial function, and the modulation of cellular signaling and gene expression [2,45,46]. Owing to its high negative charge density and flexible polymeric conformation, polyP can influence fundamental protein properties, including structural stability, functional activity, and aggregation propensity [47,48]. However, due to its nonspecific affinity for exposed, unstructured protein regions, it is essential to experimentally validate these interactions to determine their physiological relevance. In osteoblastic cells, polyP has been shown to regulate collagen folding by modulating cyclophilin B, a key mediator of this process [47].

PolyP stabilizes protein folding intermediates, acting as a molecular chaperone that prevents protein aggregation and protects cells from stress-induced damage [2,45,49]. It also accelerates amyloid fiber formation, alters fiber morphology, and can protect cells from amyloid toxicity, suggesting a role in neurodegenerative disease processes [45,49]. This dual role positions polyP as both a protective and potentially pathological modulator of protein aggregation that is particularly relevant to amyloidogenic pathways, which will be further discussed.

In bacterial models, the accumulation of polyP under stress conditions, such as thermal or oxidative stress, has traditionally been interpreted as a cytoprotective strategy. In these scenarios, polyP acts as a primordial chaperone that prevents the irreversible aggregation of partially unfolded proteins, stabilizing them in soluble β-sheet-rich conformations and facilitating their refolding by molecular chaperones such as DnaK [50,51,52,53].

PolyP also modulates liquid–liquid phase separation, influencing the formation of cellular condensates and potentially affecting gene expression [45]. This property is particularly relevant in the context of stress granules and RNA-binding proteins, some of which are implicated in neurodegenerative disorders.

### 4.2. Modulator of Amyloidogenesis

Amyloidogenic proteins share an intrinsic property to undergo conformational transitions from soluble, disordered states to β-sheet-rich structures. These conformations are highly prone to aggregation, leading to a structural hierarchy that includes oligomers, protofibrils, and ultimately mature fibrils, which are characterized by a cross-β architecture [54,55].

A critical aspect of the pathogenicity associated with these proteins is the ability of mature fibrils to release oligomers and protofibrils—intermediate species that propagate and promote cellular toxicity. These soluble forms disrupt several cellular processes, including ionic homeostasis and membrane integrity, contributing to synaptic dysfunction and neuronal death [56,57,58].

Within this context, polyP has emerged as a key modulator of amyloidogenic processes. The current findings demonstrate that polyP interacts with β-sheet-rich conformational intermediates, stabilizing them and promoting both the nucleation and elongation of amyloid fibrils. This pro-amyloidogenic role has been observed in both functional amyloids, such as the bacterial CsgA protein, and pathological amyloids implicated in neurodegenerative diseases, including Aβ, α-synuclein, and tau [46,50,51]. PolyP has been shown to significantly accelerate the onset of activity of tau fibril formation, shortening the transition time from several months to just a few hours under experimental conditions. This effect is directly correlated with the polyP chain length, with longer chains exhibiting greater capacity to promote both nucleation and fibril elongation [46].

Beyond its pro-aggregating role, polyP also modifies the morphology and stability of mature fibrils. Specifically, its presence reduces susceptibility to proteolysis and decreases the release of oligomeric species, potentially limiting the propagation of toxic intermediates [46]. Functional assays in neuronal cultures have demonstrated that fibrils formed in the presence of polyP lose their cytotoxic effects, suggesting a potentially protective or regulatory function of polyP under physiological conditions [46].

Nevertheless, critical questions remain regarding the precise mechanisms by which polyP modulates amyloidogenesis. Specifically, a more accurate characterization is needed to determine the timing of polyP interaction with forming aggregates and how this interaction alters the structural dynamics of fibrils. It is also essential to elucidate the impact of polyP in physiological environments, particularly within the brain, and to assess its feasibility as a therapeutic target in neurodegenerative diseases. Advances in super-resolution microscopy and molecular labeling strategies hold promise for clarifying these mechanisms in future studies [1,46,59].

Figure 5 summarizes the described scaffolding functions of polyP.

### 4.3. Modulation of the mPTP

Studies in primary rat hepatocytes have demonstrated that polyP interacts with the mPTP, located in the inner mitochondrial membrane, as well as with ion channels such as Trpm2 and TRPA1 [60]. In murine cardiomyocytes, the effect of polyP on mPTP activity has been shown to depend directly on the polymer chain length, with its reduction attenuating Ca^2+^-induced pore opening. This finding suggests the existence of an autocrine regulatory mechanism involving the Ca^2+^–polyP axis [61].

Activation of the mPTP disrupts mitochondrial homeostasis by facilitating massive calcium influx, promoting mitochondrial swelling, and favoring apoptotic pathways [30]. Moreover, mitochondrial polyP levels are tightly coupled to the metabolic state of the cell: they increase in response to respiratory substrates and decrease when mitochondrial inhibitors or ATP synthase blockers are applied [5,7,28,30,31]. The depletion of mitochondrial polyP results in impaired oxidative phosphorylation, a compensatory upregulation of glycolysis, and mitochondrial fragmentation, indicating its crucial role in regulating the metabolic balance between these energy pathways.

These functions, which are closely linked to the maintenance of the mitochondrial membrane potential and redox balance, are essential for cell survival and viability. Emerging evidence indicates that polyP acts as an integrative regulator of these pathways, modulating both physiological events and stress response mechanisms within mitochondria [62].

### 4.4. Regulation of Bioenergetic Processes

Energy metabolism in mammalian cells relies on key pathways such as mitochondrial oxidative phosphorylation (OXPHOS), glycolysis, and the pentose phosphate pathway (PPP). These processes ensure efficient ATP production and cellular energy balance. Dysfunction of the electron transport chain (ETC) increases the production of reactive oxygen species (ROS), leading to oxidative stress. As a compensatory mechanism, the PPP contributes to the synthesis of NADPH, which is essential for maintaining redox homeostasis [8,44,60,61,62]. Recent findings have expanded this framework by identifying polyP as a key regulator of mitochondrial energy metabolism [6,63,64]. This linear polymer of orthophosphate residues linked by high-energy phosphoanhydride bonds structurally like those in ATP possesses a bioenergetic potential comparable to that of ATP [62,65]. Its accumulation in mitochondria suggests a multifaceted role, including calcium homeostasis, the modulation of oxidative stress, and direct involvement in energy-dependent processes such as OXPHOS and apoptosis [6,44,62].

PolyP is both synthesized and hydrolyzed by the mitochondrial F_0_F_1_-ATP synthase, indicating a bidirectional role in the cellular energy balance [29,30]. Experimental data show that mitochondrial polyP activates respiration even in the absence of ATP synthesis while inhibiting ADP-dependent respiration [29]. The levels of mitochondrial polyP vary depending on the metabolic state and availability of respiratory substrates and are reduced by inhibitors such as rotenone or uncouplers like FCCP [31]. Notably, oligomycin, a classical ATP synthase inhibitor, also blocks polyP synthesis [31]. These findings support a dynamic feedback loop in which the mitochondrial energy status regulates polyP production [31,60,61,66].

Experimental studies have shown that mitochondrial polyP depletion induces a metabolic shift from oxidative phosphorylation toward glycolysis [63]. This shift carries significant pathophysiological implications, as it has been associated with adaptive responses to hypoxic conditions, altered cell proliferation, and the progression of neurodegenerative and oncological diseases [8,60,61]. In polyP-depleted HEK293 cells, this bioenergetic switch is accompanied by enhanced glycolytic flux, increased pentose phosphate pathway (PPP) activity, and elevated levels of oxidative stress markers such as mitochondrial superoxide and intracellular hydrogen peroxide [63,67]. These cells also exhibit the upregulation of antioxidant proteins, including peroxiredoxin-1 (Prx1), superoxide dismutase-2 (SOD2), and thioredoxin (Trx), indicating compensatory responses to the redox imbalance [67].

The regulatory role of polyP in bioenergetics is further reflected in its direct interaction with ATP synthase, an enzyme capable of both synthesizing and degrading polyP in vitro, implying a feedback loop between polyP metabolism and ATP availability [63]. This is corroborated by studies demonstrating that the enzymatic depletion of polyP alters ATP production kinetics and the redox balance [1,16]. In this context, long-chain polyP species have been shown to enhance ADP-linked respiration and support mitochondrial ATP synthesis [30,66]. PolyP emerges not only as an energy buffer but also as a key signaling molecule that coordinates mitochondrial respiration, redox control, and calcium-dependent processes [68,69].

Alterations in mitochondrial polyP levels have been associated with various human diseases, including neurodegenerative disorders such as ALS and FTD, as well as hematological and cardiovascular pathologies. In this context, polyP contributes to cellular viability by modulating ROS production, supporting mitochondrial function, and maintaining intracellular signaling integrity [28].

This highlights an emerging complexity in polyP regulation, with implications for mitochondrial function and disease.

Altogether, these findings position polyP as a multifunctional metabolite in mitochondrial bioenergetics, integrating ATP metabolism, redox control, and structural regulation through chain-length-specific effects. The ongoing exploration of polyP metabolism not only enriches our understanding of mitochondrial physiology but also opens promising avenues for the development of therapeutic interventions in diseases marked by energetic dysfunction and mitochondrial instability [28,70].

### 4.5. Mitochondrial Calcium Homeostasis

PolyP plays a crucial role in the regulation of mitochondrial calcium (Ca^2+^) transport and metabolism, acting through multiple interrelated mechanisms.

From a functional perspective, polyP is essential for maintaining sustained concentrations of free calcium in the mitochondrial matrix. In HEK293 cell models engineered to overexpress MitoPPX, an enzyme responsible for polyP degradation, it was observed that the depletion of this polymer does not affect Ca^2+^ uptake via the mitochondrial calcium uniporter (MCU) or the membrane potential, but significantly reduces free Ca^2+^ concentrations in the matrix [68]. This finding highlight polyP’s role as a modulator of the mitochondrial buffering capacity through the chelation of divalent cations, primarily Ca^2+^.

Recent studies have demonstrated that polyP forms soluble complexes with calcium in the mitochondrial matrix, preventing the formation of insoluble calcium–phosphate precipitates. This mechanism allows mitochondria to maintain elevated levels of free, bioavailable calcium during periods of calcium influx, which are crucial for cellular signaling and metabolism [62,68,69]. In the absence of polyP, calcium buffering relies solely on orthophosphate, which can lead to the formation of insoluble precipitates, potentially disrupting mitochondrial function [69].

The regulation of mitochondrial calcium by polyP is critical for the activation of tricarboxylic acid (TCA) cycle dehydrogenases, which are essential for NADH production and, consequently, ATP synthesis [69]. In this way, polyP emerges as a key integrator of ionic signaling and metabolic pathways, particularly under physiological and pathological conditions.

Mitochondrial polyP levels are highly dynamic and reflect the metabolic state of the organelle. It has been proposed to act both as a reservoir of high-energy phosphoryl groups and as a direct regulator of enzymatic activity under cellular stress conditions, such as ischemia–reperfusion or heart failure [31]. The mitochondrial enzyme F_0_F_1_-ATP synthase has been identified as responsible for both the synthesis and degradation of polyP, directly implicating this complex in mitochondrial bioenergetic regulation [29,30,71]. These findings position polyP as a multifunctional regulator of mitochondrial physiology, participating in diverse processes such as ion transport, ATP synthesis, mPTP opening, and the cellular damage response. Therefore, its modulation represents a promising avenue for the development of therapeutic strategies targeting mitochondrial and neurodegenerative diseases.

PolyP’s roles in mitochondrial homeostasis and bioenergetic processes are summarized in Figure 6.

### 4.6. The Cellular Stress Response

PolyP is a key regulator of mitochondrial function, contributing to energy metabolism and proteome stability. In mammalian cells, its mitochondrial accumulation suggests an active role in proteostasis and the adaptation to cellular stress. Experimental evidence from bacterial and mammalian models has demonstrated that polyP depletion increases protein aggregation under stress conditions, indicating that this polymer functions as an essential modulator of the protein conformational state and gene expression in both physiological and pathological contexts [46,49,50,51,72,73,74].

One of its most prominent effects is its ability to positively modulate the mPTP, a key structure in the control of mitochondrial bioenergetics and integrity. It has been proposed that polyP may be part of a structural complex in the inner mitochondrial membrane, together with poly-β-hydroxybutyrate (PHB) and Ca^2+^, thereby facilitating ionic transport [69]. The importance of polyP in the regulation of mPTP has been highlighted by Seidlmayer et al. (2012), who suggested that this polymer may constitute a functional component of the pore complex [75]. Furthermore, its high co-localization with mitochondria in mammalian cells supports its involvement in maintaining the cellular energy balance [69,75]. This hypothesis is reinforced by studies demonstrating a positive correlation between polyP levels and ATP concentrations in various cell types, including fibroblasts, renal cells, and adrenal cells [15].

In bacteria, polyP exhibits chaperone-like functions, stabilizing folding intermediates and interfering with amyloid structure formation, underscoring its evolutionarily conserved role in protein protection. This effect has also been documented in human cells, where the intracellular depletion of polyP is associated with increased sensitivity to stressors, including oxidative agents and thermal fluctuations [76,77]. At the bioenergetic level, polyP is proposed to act as a dynamic reservoir of phosphoryl groups, contributing to the regulation of oxidative phosphorylation and protein translocation across mitochondrial membranes [49].

In this context, elevated levels of polyP in mammalian cell mitochondria modulate the subcellular localization of phosphorylated proteins and regulate the production of ROS, thus playing a significant role in antioxidant defenses. PolyPs influence the levels and activity of key antioxidant enzymes such as superoxide dismutase (SOD), peroxiredoxin-1 (Prx1), and thioredoxin (Trx), which are crucial for neutralizing ROS and maintaining the redox balance [2,67]. They also interact with metal ions to modulate redox reactions and stabilize proteins under stress conditions [76] while acting as protein-protective chaperones to prevent aggregation and damage [2,76]. Moreover, polyPs and their derivatives can directly scavenge free radicals, including 2,2-diphenyl-1-picrylhydrazyl (DPPH) and hydroxyl radicals [2], and they regulate the pentose phosphate pathway (PPP), promoting NADPH production for antioxidant regeneration [67]. The increased levels of SOD, Prx1, and Trx observed in polyP-depleted cells further suggest that polyP supports antioxidant enzyme systems [2,67].

The enzymatic depletion of polyP directly affects the integrity of the mitochondrial quality control system. In cellular models deficient in polyP (MitoPPX), delayed activation of the mitochondrial unfolded protein response (UPR mt) has been observed, as evidenced by increased DDIT3 expression following exposure to rotenone [78,79]. Although ATF5 is recognized as a central transcription factor in UPR mt activation, its effectiveness critically depends on its localization and abundance within mitochondria, suggesting the existence of additional regulatory mechanisms [80].

The protective role of polyP during cellular stress is further underscored in models of ischemia–reperfusion injury. During ischemia, polyP contributes to the regulation of mitochondrial calcium levels and transient opening of the mPTP, mechanisms that may prevent calcium overload and mitigate mitochondrial damage. Upon reperfusion, the depletion of polyP has been associated with increased production of ROS and elevated rates of cell death, emphasizing its importance in buffering oxidative insults during metabolic stress [61,81]. In cardiac myocytes, this protective function is particularly evident; polyP deficiency exacerbates oxidative damage and promotes cell death under ischemic and reperfusion conditions, highlighting its critical role in redox homeostasis and cell survival in energy-demanding tissues such as the heart [60,81].

Moreover, reduced mitochondrial polyP levels are correlated with decreased expression of SIRT3, a key sirtuin involved in energy metabolism regulation and mitochondrial quality control. This reduction may impair the cellular capacity to adapt to proteotoxic stress [82,83]. Nevertheless, under stress conditions, SIRT3 levels remain stable in both wild-type and MitoPPX cells, suggesting the activation of compensatory mechanisms. Concurrently, overexpression of SOD2, a mitochondrial antioxidant enzyme whose induction is associated with increased ROS levels due to oxidative phosphorylation dysfunction, has been observed [63,67,84].

On the other hand, the expression of the chaperones Hsp60 and Hsp10, encoded by HSPD1 and HSPE1, respectively, remains unchanged following polyP depletion, as do ATF5 levels, indicating that the protective effect of polyP on mitochondrial proteostasis extends beyond these factors. Similarly, reduced transcription of PRKN under stress conditions in MitoPPX cells, without detectable changes in protein levels, and the absence of variations in genes associated with mitochondrial dynamics such as DNM1L and TOMM20 support the notion that polyP modulates specific components of the mitochondrial proteostasis system [77].

### 4.7. Role of PolyP in Pi Regulation

PolyP not only serves as a Pi storage form but also buffers intracellular Pi fluctuations, supporting cellular demands during biosynthetic events or stress conditions such as nutrient scarcity or increased DNA synthesis [85,86]. Although the mitochondrial concentrations of polyP are relatively low, mitochondria possess specific transport systems—primarily a specialized H^+^/Pi symporter—that allow efficient uptake of cytosolic Pi [87]. This functional interplay between cytosolic polyP pools and mitochondrial Pi transport highlights the role of polyP in maintaining phosphate homeostasis and supporting mitochondrial metabolism.

Moreover, the regulation of polyP synthesis and Pi transport is modulated by inositol polyphosphates (InsPs), which interact with SPX domains to communicate the cellular Pi status, enabling adaptive responses to fluctuating phosphate levels [88,89]. In this context, polyP is proposed to act as an indirect modulator of mitochondrial redox homeostasis by regulating Pi availability. This highlights its potential relevance in maintaining the cellular oxidative balance and preventing damage associated with oxidative stress. Indeed, excessive intracellular Pi can disrupt mitochondrial metabolism, promoting elevated ROS production, which may trigger mitochondrial dysfunction, oxidative stress, and programmed cell death—particularly in susceptible cell types such as pancreatic and insulin-secreting cells [90]. Similarly, inhibition of the mitochondrial Pi transporter has been shown to effectively reduce superoxide production and prevent pathological processes such as phosphate-induced vascular calcification [91]. Together, these observations suggest that polyP’s ability to manage intracellular Pi availability not only supports metabolic needs but also plays protective roles in the redox balance, energy metabolism, and stress adaptation [12,88,89,92].

### 4.8. PolyP in in Astroglial Signaling and Neuronal Homeostasis

PolyP is present at micromolar concentrations in the mammalian brain, where it acts as a mediator of astroglial intercellular communication [32,93]. In vivo studies have demonstrated that polyP modulates critical homeostatic functions in the brainstem, including increased respiratory activity, central sympathetic outflow, and arterial blood pressure [32]. Moreover, intracellular polyP levels fluctuate according to the cellular metabolic status and age, suggesting a role in the modulation of neuronal homeostasis. The concentration of polyP in the CNS ranges between 25 and 120 μM [32]. Fluorescence-based methods using DAPI have visualized its intracellular distribution in astrocytes, showing localization in the nucleus, lysosomes, mitochondria, and cytoplasm, which suggests distinct functional roles in each compartment. Its release is regulated by intracellular calcium fluctuations, as observed following ionomycin stimulation [32].

PolyP induces increases in the intracellular calcium concentration ([Ca^2+^]^c^) in astrocytes, and to a lesser extent in neurons, in a manner dependent on the polymer length and concentration [32]. This effect is more pronounced on astrocytes derived from the midbrain, hippocampus, and cortex. The length of the polyP chain determines its functional effects: short- and medium-chain polymers participate in intracellular signaling and mitochondrial metabolism, whereas long-chain polyP promotes mitochondrial depolarization and opening of the permeability transition pore (PTP), leading to astrocytic cell death. This cytotoxicity can be attenuated by PTP inhibitors such as cyclosporin A [93,94].

In terms of signaling, polyP activates purinergic P2Y1 receptors, promoting phospholipase C (PLC) activation and the mobilization of Ca^2+^ from intracellular stores [32,94]. It also modulates thymocyte function via the activation of purinergic P2X receptors [95,96]. Regarding the subcellular localization, polyP has been identified in lysosomes of both astrocytes and neuronal synaptosomes, where it may regulate neuronal excitability through voltage-dependent channels [97]. However, only a subset of these lysosomes undergo fusion with the plasma membrane in response to stimuli, indicating a tightly regulated release mechanism.

Recent studies have shown that mitochondria are capable of synthesizing polyP, and that approximately 40% of intracellular polyP in astrocytes localizes to this organelle [31,93,98]. This mitochondrial localization is dynamic and energy state-dependent [31,62]. Additionally, polyP modulates the activity of TRPA1 and TRPM8 ion channels, indicating a role in the regulation of sensory signaling [32].

In cultured astrocytes, polyP is predominantly found in vesicles expressing the vesicular nucleotide transporter (VNUT), but not in those expressing the vesicular glutamate transporter VGLUT2, suggesting a selective role in purinergic signaling via molecules such as ATP [93]. The release of polyP from VNUT-positive vesicles may trigger intracellular signaling cascades in astrocytes, and the dysregulation of this process could be implicated in the pathophysiology of neurodegenerative and neuropsychiatric disorders [98,99,100].

Beyond its effects on astrocytes, polyP also plays a central role in maintaining neuronal homeostasis through multiple mechanisms. Inositol polyphosphates, including IP_3_ and IP_4_, are key modulators of calcium signaling in neurons, mobilizing calcium from intracellular stores and regulating potassium currents, thereby influencing excitability and synaptic transmission [5,101,102,103]. Notably, short-chain polyP species are particularly effective at reducing neurotoxic calcium elevations and promoting neuronal survival, whereas long-chain forms can be detrimental [102,103]. Within mitochondria, polyP supports bioenergetic stability, and its depletion leads to mitochondrial dysfunction and energy deficits, like those observed in aging and neurodegenerative diseases [10]. In pathological contexts such as Alzheimer’s disease (AD), polyP has been shown to counteract β-amyloid-induced reductions in ATP levels, suggesting a neuroprotective role [104]. Additionally, polyP functions as a molecular chaperone, preventing stress-induced protein misfolding and aggregation, and it may also act as a gliotransmitter, facilitating communication between astrocytes and neurons [2,5,32,103]. Together, these findings underscore the multifaceted role of polyP in neuronal survival, mitochondrial integrity, and synaptic communication.

PolyP and inositol polyphosphates are emerging as key regulators of neural signaling under physiological conditions. Similar to astrocytes, polyP in neurons is abundant in mitochondria, and it modulates the cellular energy balance by influencing AMPK signaling, a master regulator of mitochondrial homeostasis [105]. Through this mechanism, polyP couples the metabolic state to neuronal function. Extracellularly, polyP and nucleotides such as ATP, ADP, and diadenosine polyphosphates act as extracellular signaling molecules whose breakdown by ecto-nucleotidases is essential for regulating synaptic transmission, neural cell survival, and differentiation [106]. Inositol polyphosphates, including IP3 and IP4, act as second messengers in both the cytoplasm and nucleus, influencing diverse pathways that govern neural tube development and embryogenesis [107]. Additionally, the phosphoinositide signaling cascade mediated by polyphosphoinositides in neural membranes generates diacylglycerol and inositol trisphosphate, which activate protein kinase C and mobilize intracellular calcium to regulate neural responses like motility and contractility [108].

However, disruptions in polyP metabolism can contribute to neural dysfunction and disease. Abnormal polyP levels may impair mitochondrial bioenergetics and deregulate AMPK activity, weakening neuronal metabolic adaptability [105]. Similarly, defects in the synthesis or signaling of inositol polyphosphates can lead to disturbed calcium dynamics and developmental defects, as seen in models of impaired neurogenesis [107]. These alterations compromise signaling fidelity and cellular resilience, potentially promoting oxidative stress, synaptic dysfunction, and neuronal death in pathological contexts. The role of polyP dysregulation in these disease processes will be discussed in detail in the following section.

## 5. PolyP and CNS Disorders: A New Perspective on Neurodegenerative Mechanisms

In the nervous system, polyP plays an essential role in intercellular communication and protein stability. It has been shown to function as a neuromodulator through the activation of purinergic receptors in astrocytes and displays protective properties by preventing the aggregation of amyloidogenic proteins—a key pathological event in neurodegenerative disorders such as Alzheimer’s disease (AD) and Parkinson’s disease (PD) [2]. As previously discussed, at the mitochondrial level, polyP regulates calcium homeostasis and protects against mitochondrial dysfunction, processes directly linked to neuronal death in these conditions [30,44].

The impact of polyP on immune responses is also relevant to neurodegeneration. The instability of mitochondrial proteins has been linked to various human diseases, although the precise mechanisms are still unclear [62,70,109]. Its ability to modulate macrophage and neutrophil activation has been demonstrated, contributing to chronic inflammation, a pathogenic hallmark of neurodegenerative diseases [110]. It has been proposed that its interaction with cellular proteins may be mediated through post-translational polyphosphorylation, potentially explaining its remarkable functional versatility [111]. Nevertheless, the accurate quantification of its concentration and intracellular distribution remains a significant technical challenge [112].

PolyP supports mitochondrial bioenergetics by modulating ATP synthase activity, a function disrupted in neurodegenerative conditions like AD [70]. Its regulatory role may help restore the neuronal energy balance and mitigate mitochondrial dysfunction [113]. The partial colocalization of polyP with lysosomal markers such as CD63-mKate and its association with the cellular polyP content based on aging and the metabolic status reinforce its importance in maintaining cellular homeostasis [32]. Furthermore, deficiency of LRRK2—a kinase associated with PD—markedly reduces the proportion of lysosomes containing polyP, suggesting a link between polyP dysregulation and neurodegenerative processes [114].

Beyond its intracellular roles, polyP is also secreted by various cell types, including procoagulant platelets, activated mast cells, and, notably, astrocytes in the CNS. Astrocyte-derived polyP plays a crucial role in neurotransmission by stimulating neuronal activity and inducing calcium-mediated intracellular signaling [115]. This extracellular function of polyP underscores its growing relevance not only in normal brain physiology but also in the pathogenesis of several neurodegenerative disorders [8], such as ALS, FTD, AD, and PD [116]. Recently, elevated polyP levels have been identified in astrocytes and cerebrospinal fluid from ALS and FTD patients, as well as in corresponding murine models. Experimental studies have shown that inhibiting astrocyte-secreted polyP confers neuroprotection to motor neurons, suggesting a critical role for this polymer in neurodegenerative mechanisms and its potential as a therapeutic target [117]. Additionally, polyP has demonstrated neuroprotective properties in protein misfolding disorders. It has been reported to mitigate the progression of AD, PD, and Huntington’s disease, possibly by interfering with pathological aggregate formation [92].

PolyP is both secreted and internalized by neuronal cells, allowing it to participate actively in intracellular signaling and the regulation of neuronal excitability. Previous studies have shown that polyP modulates voltage-gated channel activity, thereby altering neuronal excitability, and it acts as a signaling molecule in intercellular communication in the mammalian brain [32,97].

Studies in various neuronal cell cultures, such as differentiated human neuroblastoma cells and PC12 cell lines, have shown that the presence of polyP during amyloid fibril formation can abolish their cytotoxicity. A similar cytoprotective effect was observed when polyP was added shortly before exposure to preformed amyloid fibrils, indicating a direct role of the polymer in neutralizing the toxicity associated with amyloid aggregates [16,46]. The exact mechanisms by which polyP protects cells from amyloid-induced toxicity remain incompletely understood. The prevailing model of amyloid toxicity proposes that conformational intermediates, such as oligomers and protofibrils, rather than mature fibrils, are primarily responsible for membrane disruption, calcium homeostasis loss, and aberrant Ca^2+^ signaling, culminating in cell death [118]. Within this framework, it has been proposed that polyP may reduce the effective concentration of toxic oligomers by accelerating fibril formation, suppressing oligomer dissociation, or inhibiting primary nucleation. Alternatively, polyP-associated fibrils may evade recognition by the cellular receptors responsible for aggregate uptake, such as heparan sulfate proteoglycans [119], or polyP may facilitate the sequestration or turnover of fibrils within inert cellular compartments.

Furthermore, mitochondrial polyP depletion has been shown to inhibit the opening of the mPTP, thereby interfering with the β-amyloid-induced apoptotic cascade. This highlights a potential indirect mechanism by which polyP may modulate neuronal vulnerability to toxic aggregates [44]. Finally, considering that a decline in cerebral polyP levels has been reported with aging in both rat tissues and human cells, it has been hypothesized that such a reduction may contribute to the onset or progression of amyloid-related neurodegenerative diseases [120].

### 5.1. PolyP in Amyotrophic Lateral Sclerosis (ALS) and Frontotemporal Dementia (FTD)

ALS and FTD represent a clinical spectrum of neurodegenerative disorders that share overlapping pathogenic mechanisms, including the progressive degeneration of motor neurons (MNs) and cortical neurons, disruption of RNA and protein homeostasis, and the involvement of mutations in genes such as TARDBP, SOD1, C9ORF72, and VCP [116,121,122].

These diseases involve significant glial dysfunction, especially pathological astrocytes, which can induce motor neuron toxicity through non-cell-autonomous mechanisms. Mitochondrial dysfunction and neuroinflammation are also key processes driving disease progression in both ALS and FTD. Several studies have demonstrated that mutated or dysfunctional astrocytes can induce toxicity in MNs through non-cell-autonomous mechanisms, even in the absence of direct genetic mutations in the motor neurons themselves [9,10,116,123].

Human cellular models based on astrocytes derived from induced pluripotent stem cells (iPSCs) carrying mutations in TARDBP (such as TDP-43 A90V), SOD1, or C9ORF72 have revealed a moderately reactive phenotype, with a mature morphology and the expression of markers such as GFAP, S100β, and ALDH1L1. Despite this phenotypic maturity, these astrocytes exhibit functional deficiencies, particularly in synaptogenesis, and a reduced capacity for neuronal support [124,125]. These models faithfully replicate characteristics observed in patients and have allowed the study of both the autonomous and non-autonomous contributions of astrocytes to neurodegeneration [126,127].

Recent studies have demonstrated that polyP contributes to mitochondrial dysfunction in reactive astrocytes, where its intracellular accumulation and subsequent release are associated with bioenergetic alterations and the emission of damage signals that increase neuronal vulnerability. Although the underlying molecular mechanisms are not yet fully understood, it has been proposed that polyP interferes with mitochondrial calcium metabolism and promotes the opening of the mPTP, potentially facilitating the release of pro-apoptotic signals [15]. In this context, the neutralization of polyP in astrocyte-conditioned media has been shown to reduce neuronal death, suggesting its potential as a therapeutic target. Furthermore, elevated levels of polyP have been reported in the cerebrospinal fluid of patients with mutations in the TARDBP gene, reinforcing its pathogenic role in vivo [122].

One of the most relevant findings in these models has been the excessive release of polyP by mutant astrocytes, a phenomenon identified in both human and murine cell lines. This polyP has been shown to exert toxic effects on motor neurons, positioning it as a novel mediator of glial toxicity [9].

The experimental depletion of polyP has been shown to significantly reduce the toxicity induced by mutant astrocytes, validating its causal role in neuronal degeneration [9]. Furthermore, excessive polyP production correlates with a dysfunctional astrocytic profile, independent of interactions with neurons or microglia. This pathogenic behavior has been observed even in the absence of external inflammatory signals, indicating a cell-autonomous origin [126,127].

In addition to polyP, other mechanisms have been proposed to explain astrocytic toxicity. The dysfunctional release of microRNAs, such as miR-146a, through extracellular vesicles has been evidenced, with potential degenerative effects on neuronal networks [128,129]. The involvement of Cx43 hemichannels has also been documented, where aberrant activation favors the release of additional toxic signals [123].

These findings position polyP as an emerging biomarker and a potential therapeutic target in ALS and FTD. The evidence supports the hypothesis that iPSC-derived astrocytes with mutations in genes associated with these diseases actively contribute to neurodegeneration through both autonomous and non-autonomous mechanisms. Therefore, strategies aimed at modulating polyP production and release could represent novel approaches for the development of targeted therapies.

### 5.2. Involvement of PolyP in Alzheimer’s Disease (AD)

AD is a progressive neurodegenerative disorder characterized by the accumulation of β-amyloid (Aβ) plaques and neurofibrillary tangles composed of the hyperphosphorylated Tau protein, which lead to synaptic and neuronal degeneration [130,131,132]. It has been proposed that Aβ-induced dysfunction precedes Tau alterations and constitutes a key pathogenic axis in the progression of the disease [133,134]. Additionally, Aβ misfolding activates neuronal apoptotic pathways [135,136], prompting the development of therapies aimed at preventing its misfolding [137].

In parallel, mitochondrial dysfunction has been identified as an early event in AD associated with oxidative stress and impaired ATP production, suggesting a causal role in the neurodegenerative process. Mitochondria actively participate in energy metabolism, redox homeostasis, and the modulation of inflammatory responses, positioning them as a central component in the pathophysiology of the disease.

PolyP has gained attention for its essential role in mitochondrial bioenergetics. Its absence has been linked to functional alterations like those observed in neurodegenerative diseases, including AD. One proposed pathway involves the participation of polyP in the activation of AMP-activated protein kinase (AMPK), a master sensor of the cellular energy status. The disruption of the mitochondrial polyP content has also been associated with increased vulnerability to oxidative damage, defects in mitochondrial biogenesis, and a reduced cellular adaptive capacity to stress. Further studies have shown that polyP in its amorphous formulation as calcium microparticles (Ca-polyP-MP) exerts a significant neuroprotective effect. It has been shown that a pre-incubation of neuronal cells with Ca-polyP-MP restores ATP levels after exposure to Aβ25–35, mitigating the resulting neurotoxicity [104]. Consistently, intracellular polyP levels decrease with age, which could increase neuronal susceptibility to degenerative processes such as AD. Cell viability assays using MTT in PC12 cells have shown that polyP exhibits no cytotoxicity at concentrations up to 30 μg/mL, and its neuroprotective efficacy depends on its structural form, being more potent in its amorphous formulation [138].

In addition to its bioenergetic functions, polyP may directly interact with key proteins implicated in AD. For example, there is evidence that this polymer can conformationally stabilize the Aβ peptide and interfere with its pathological aggregation, thereby reducing the formation of senile plaques [2].

This direct modulation of protein aggregation, together with polyP’s role in preserving cellular energy and enhancing extracellular chaperone function, such as that of clusterin, positions it as a multifaceted regulator of AD progression. Moreover, polyP’s potential to cross the blood–brain barrier, especially when this barrier is compromised in later stages of the disease, further broadens its therapeutic promise [139]. Taken together, these diverse actions underscore polyP as a compelling target for future therapeutic strategies aimed at slowing or preventing AD.

### 5.3. PolyP and Neuronal Senescence in Brain Aging

Neuronal senescence represents an irreversible state of cell cycle arrest that involves profound morphological, functional, and transcriptomic alterations, including the expression of a senescence-associated secretory phenotype (SASP) that contributes to a proinflammatory and neurotoxic environment in the CNS [140,141]. In this context, mitochondrial dysfunction has been identified as a key driver of neuronal senescence, particularly in neurodegenerative diseases such as Alzheimer’s disease and Parkinson’s disease, through mechanisms including impaired oxidative phosphorylation, increased ROS levels, and a loss of mitochondrial membrane potential [142,143,144,145].

Furthermore, dietary restriction and fasting have been shown to have beneficial effects on modulating longevity and preventing pathological aging through the activation of protective metabolic pathways such as AMPK, which regulates energy metabolism and favors cellular homeostasis by restoring autophagic flux [146,147,148]. In this framework, polyP emerges as an essential regulator of bioenergetic processes [68,149]. Tagliafico et al. (2024), using differentiated SH-SY5Y neuroblastoma cell models modified to deplete mitochondrial polyP (MitoPPX), as well as mouse brains subjected to short-term starvation (STS), demonstrated that while STS restores polyP levels and activates the AMPK pathway, it fails to reverse the senescent phenotype in the absence of polyP [150]. MitoPPX cells exhibited classical senescence markers such as β-galactosidase activation, autophagic dysfunction, and a reduced ATP/ADP ratio, even under energy restriction conditions. Proteomic analyses further revealed alterations in critical mitochondrial pathways and the hyperactivation of AMPK accompanied by a decrease in CREB transcription factor phosphorylation, suggesting a disruption in bioenergetic signaling as an underlying mechanism [150].

These findings position mitochondrial polyP as a key functional node in regulating the cellular response to energy stress and preventing neuronal senescence induced by mitochondrial dysfunction. In this sense, the modulation of polyP metabolism emerges as a potential therapeutic strategy to slow down the neurological deterioration associated with aging [151].

From a therapeutic perspective, polyP modulation represents a promising strategy for the treatment of neurodegenerative diseases. Recent studies suggest that its exogenous administration could mitigate the toxicity of amyloid protofibrils and prevent mitochondrial damage induced by calcium overload, which could have implications in Alzheimer’s disease, Parkinson’s disease, and Huntington’s disease [112]. However, it is crucial to further elucidate its molecular mechanisms for eventual clinical application.

### 5.4. Modulation of Neuroinflammation

Neuroinflammation is a crucial component in the progression of AD and PD [152,153,154,155,156,157,158], PolyP, released by activated platelets [159], activates factor XII, promoting an exacerbated inflammatory response [36,160,161,162]. Furthermore, in human endothelial cells, polyP binds to proinflammatory nuclear proteins such as histone H4 and HMGB1, amplifying the inflammatory responses mediated by RAGE and P2Y1 receptors [94].

### 5.5. PolyP and Neuronal Apoptosis

Neuronal apoptosis, which is predominantly mediated by the mitochondrial pathway, is a central mechanism of cell loss in AD and PD [163]. PolyP has been proposed as a structural component of the mPTP alongside polyhydroxybutyrate and the C subunit of ATP synthase [164], promoting pore opening and caspase activation under certain conditions. The length of polyP is determinant: long chains (polyP120) induce caspase-3 activation and apoptosis, while shorter or medium chains have a lesser effect [5].

The roles of polyp in CNS are summarized in Figure 7.

## 6. Therapeutic Potential in Neurological Disorders

PolyP has been implicated in mitochondrial protective mechanisms against neurotoxic agents. The exogenous administration of polyP has been shown to reverse the bioenergetic damage induced by β-amyloid peptide fragments, suggesting its therapeutic potential in neurodegenerative diseases characterized by mitochondrial dysfunction [104].

Its involvement in regulating cell proliferation, inflammation, and the activity of the inositol polyphosphate multikinase (IPMK) highlights its role in energy metabolism and its interaction with critical pathways such as mTOR and AMPK, which are dysfunctional in various neurological pathologies [105].

In this context, polyP has been demonstrated to activate the mTOR pathway, a central node in the regulation of cell growth and proliferation, which may relate to its ability to restore ATP levels under stress conditions [75,81,165,166,167]. Similarly, in the oncological context, polyP has been implicated in cell proliferation via the mTOR signaling pathway and in the modulation of chemotherapy responses in brain tumors [168].

PolyP regulates calcium homeostasis and the redox balance. Its interaction with calcium signaling, in turn modulating AMPK, suggests that it could be crucial in the cellular stress response and the progression of neurodegenerative diseases such as ALS and FTD [9]. The accumulated evidence suggests that polyP could not only serve as a biomarker in pathologies such as ALS and FTD but also as a therapeutic target in conditions characterized by bioenergetic dysfunction, such as neurodegenerative disorders, COVID-19, and cancer [4,8]. Recent findings reveal a functional relationship between polyP and AMPK activation, possibly mediated by free phosphate. Changes in polyP levels directly influence AMPK activity and vice versa, suggesting a bidirectional interaction. These results position polyP as a key modulator of cellular physiology and a potential therapeutic target in diseases characterized by bioenergetic disturbances [105].

The ability of polyP to stabilize conformational intermediates rich in β-sheets has sparked interest in its potential involvement in the formation or prevention of amyloid aggregates, protein structures characteristic of neurodegenerative diseases such as Alzheimer’s disease and Parkinson’s disease [54,169]. Experimental evidence has shown that polyP reduces cytotoxicity induced by the Aβ1–42 protein in neuroblastoma cells and alleviates paralysis in transgenic *Caenorhabditis elegans* models [46]. Additionally, studies in rodents have shown a progressive decrease in brain levels of polyP with aging, which could be related to increased susceptibility to the development of amyloidogenic diseases [120].

Finally, the inhibition of polyP has shown protective effects on thrombotic events in murine models, opening new therapeutic possibilities to treat the thrombotic complications of neurological diseases [5,30,32,44,93,97,98,114].

Collectively, these findings highlight polyP as a multifunctional player in protein homeostasis, with implications ranging from fundamental cellular biology to the pathogenesis of neurodegenerative and viral diseases, positioning it as a promising therapeutic target [2,50].

## 7. Conclusions

PolyP is emerging as a key modulator of mitochondrial bioenergetics and neuroprotective processes in the CNS. Its involvement in coagulation, inflammation, calcium homeostasis, oxidative stress regulation, and the modulation of amyloidogenesis positions it as a central player in the pathophysiology of neurodegenerative diseases. Despite recent advances, critical questions remain regarding its metabolism in mammals, underscoring the need for further investigation. PolyP represents a promising therapeutic target for restoring cellular bioenergetics and mitigating the progression of neurodegenerative disorders.

Future research should focus on the detailed characterization of the biosynthetic and degradative pathways of polyP in mammalian cells, along with an analysis of its spatial and temporal dynamics under both physiological and pathological conditions. Investigating its interactions with key proteins involved in mitochondrial homeostasis and amyloidogenic processes will open new avenues for therapeutic applications. Moreover, the development of specific pharmacological strategies to modulate polyP levels in a controlled manner is essential for preserving neuronal function and counteracting the progression of CNS disorders. The integration of advanced technologies, such as super-resolution microscopy and targeted proteomics, will be pivotal in consolidating these advancements.

Although still in the early stages, exploring the modulation of polyP represents a promising therapeutic approach to reduce neuronal damage in pathological conditions. Further research is needed to investigate the biochemistry of polyP and its capacity to influence mitochondrial functions or neuroinflammatory responses. Nonetheless, this line of research could lay the groundwork for innovative strategies in translational neuroscience.

## Figures and Tables

**Figure 1 biomolecules-15-01060-f001:**
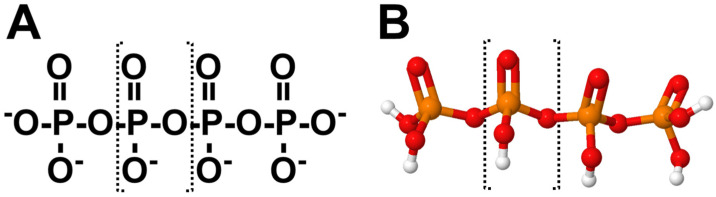
Structure of polyP. (**A**) Schematic representation of linear polyP, consisting of repeating orthophosphate (Pi) units connected by high-energy phosphoanhydride bonds (a Pi unit is shown between brackets). The figure represents a tetrapolyphosphate, but they can have from 3 to hundreds of Pi units. (**B**) Tridimensional representation of the same polyP molecule in panel A. The image was made with the web-based tool “DIY-molecules”, setting a spatial optimization of 900 steps [11]. The file with the structural information of the molecule shown in panel 1B can be downloaded as Appendix A (in .mol format).

**Figure 2 biomolecules-15-01060-f002:**
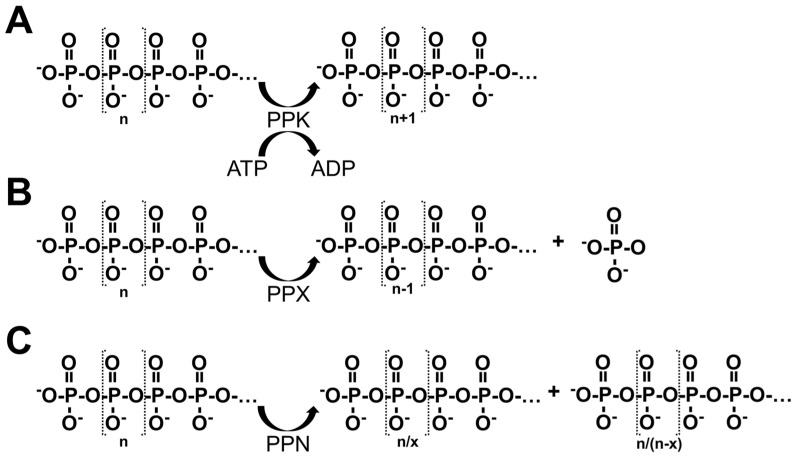
Metabolism of polyP. (**A**) Polyphosphate kinases (PPKs) catalyzes the transfer of terminal phosphate groups from ATP molecules to elongate the polyP chain. (**B**) Exopolyphosphatases (PPXs) sequentially hydrolyze phosphate residues from the terminal end of the polymer. (**C**) Endopolyphosphatases (PPNs) degrade polyP from the inner part of its chain.

**Figure 3 biomolecules-15-01060-f003:**
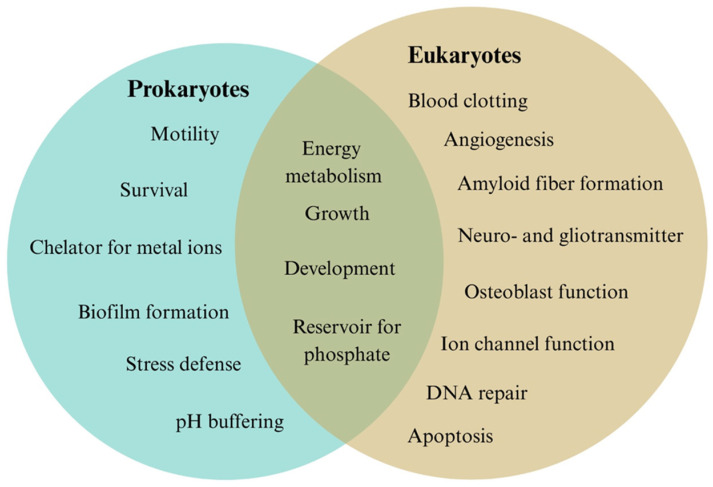
Biological functions of polyP in prokaryotic and eukaryotic organisms. The diagram shows the functions of polyP in prokaryotes and eukaryotes. PolyP is involved in key processes such as energy metabolism, growth, phosphate storage, cellular defense, blood coagulation, DNA repair, and neurotransmission, highlighting its evolutionarily conserved role.

**Figure 4 biomolecules-15-01060-f004:**
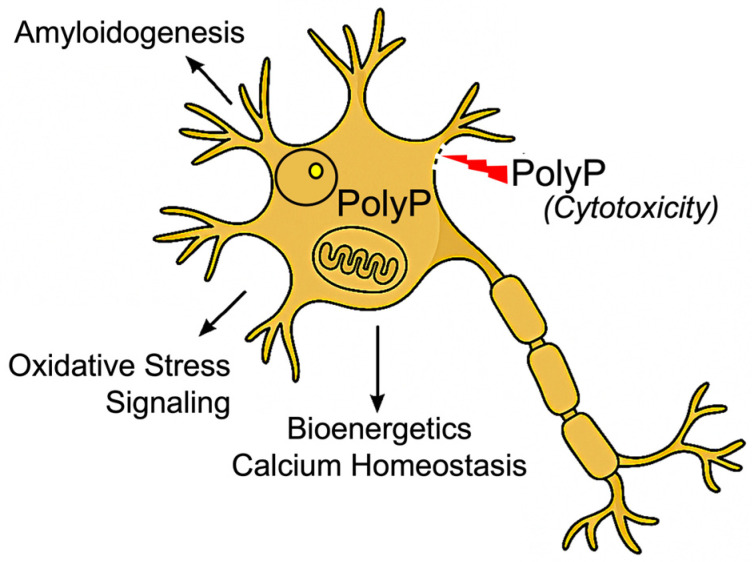
Multifaceted functions of polyP in neurons and its role in neurodegeneration. Intracellularly, polyP regulates mitochondrial bioenergetics, modulates the mitochondrial permeability transition pore (mPTP), reduces oxidative stress, and stabilizes conformational intermediates related to amyloidogenesis, thereby contributing to neuroprotection. Additionally, it participates in intracellular signaling and inflammatory processes that affect synaptic plasticity. Extracellularly, the excessive release of polyP by mutated astrocytes and activated mast cells triggers a pathological cycle of neuronal hyperexcitability, neuroinflammation, and motoneuron death, which is central to the progression of diseases such as ALS and FTD. These mechanisms position polyP as an emerging therapeutic target in CNS pathologies.

**Figure 5 biomolecules-15-01060-f005:**
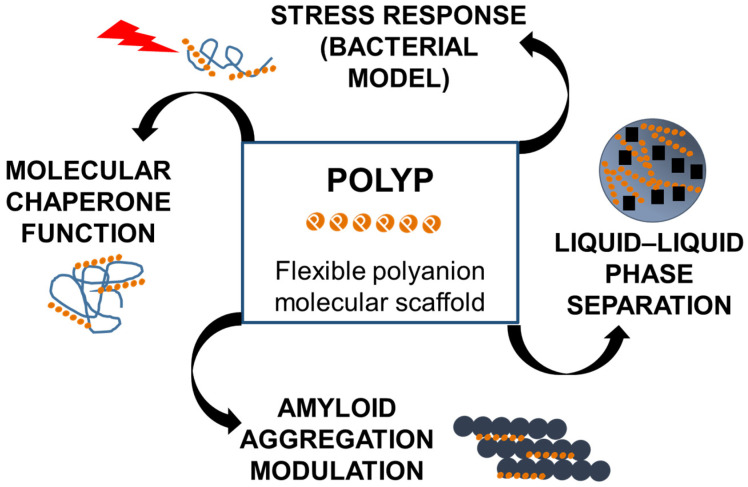
Protein scaffolding functions of polyP. This polyanion stabilizes folding intermediates and prevents protein aggregation. It protects against amyloid toxicity and promotes amyloid fiber formation with an altered morphology. In bacteria, polyP prevents irreversible aggregation under heat or oxidative stress. In addition, polyP modulates the formation of cellular condensates, promoting the formation of non-membranous granules.

**Figure 6 biomolecules-15-01060-f006:**
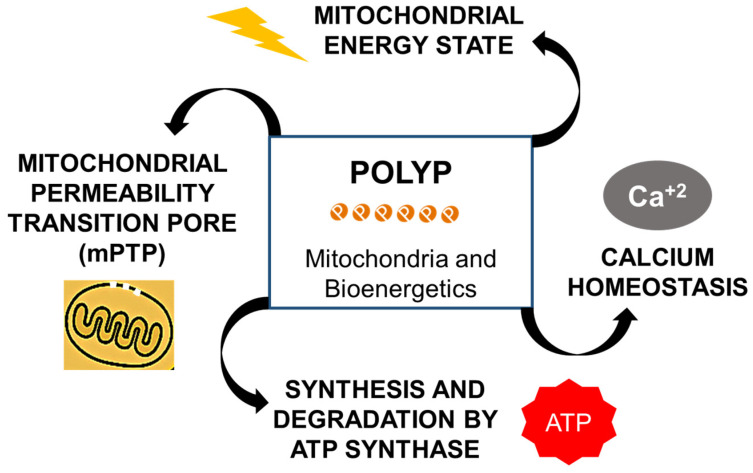
PolyP’s roles in mitochondrial homeostasis and bioenergetic processes. PolyP positively modulates the mPTP, a key structure in the control of mitochondrial bioenergetics and integrity. PolyP also maintains free mitochondrial Ca^2+^, avoiding the formation of insoluble complexes. In mitochondria, polyP is synthesized by F_0_F_1_-ATP synthase, and it can activate respiration without ATP synthesis. Finally, polyP is directly involved in energy-dependent processes such as OXPHOS and apoptosis.

**Figure 7 biomolecules-15-01060-f007:**
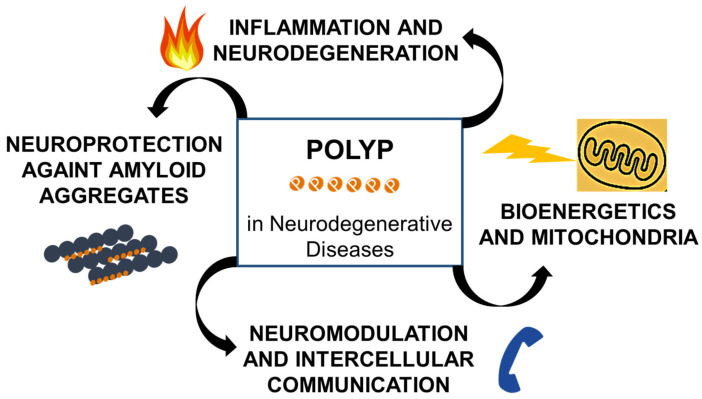
PolyP in CNS disorders. PolyP has implications in the pathogenesis of AD, PD, ALS, and FTD, such as neuromodulation, where it activates purinergic receptors in astrocytes; protein stabilization, where it prevents amyloid aggregation; mitochondrial maintenance, where it regulates calcium homeostasis, ATP synthase, and the redox balance; and the modulation of neuroinflammation, where it activates factor XII and binds to nuclear proteins to amplify inflammation via RAGE/P2Y1. In addition, dysfunctional astrocytes secrete toxic levels of polyP.

## Data Availability

Not applicable.

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
