# Peer review of "Inorganic Polyphosphate: An Emerging Regulator of Neuronal Bioenergetics and Its Implications in Neuroprotection"

_biomolecules, 2025, doi:10.3390/biom15081060_

Round 1

Reviewer 1 Report

Comments and Suggestions for Authors

The authors present a detailed review of the role inorganic polyphospate plays in cellular procesess with a focus on mammalian tissues. They discuss the role of polyP in mitochondrial function, and its potential involvement in several neurodegenerative diseases, like Alzheimer's disease, amyotrophic lateral sclerosis, and frontotemporal dementia. The paper has a total of 10 chapters, the main chapters dealing with mechanisms of synthesis, degradation and distribution of polyP; multifunctional nature of polyP; regulation of mitochondrial function; and polyP in astroglial signaling and neuronal homeostasis. Additionally, the perspective on its therapeutic potential in neurological disorders is given.

The review is comprehensive, the writting style is clear and easy to read. There are 170+ cited references which seem relevant and with only few self-citations. But there are several issues that would need to be addressed prior to its publication.

The main problem with the paper in current form, in my opinion, is the lack of representative images and tables. There is a lot of printed text and this is not reader-friendly. For such a long and detailed study it should not be a problem to add a few explanatory images and tables. This would really benefit the reader.

There are also some other issues that need to be adressed:

- The abstract is too long and reads more like an Introduction than an Abstract. Make it shorter and focus on what is the highlight of this review, what has not yet been reviewed previously.

- Add a chemical scheme of polyP and a few of the possible structures it can be found in.

- Several subchapters feel like beeing independently written by different authors without anybody reading the text afterwards and making it a homogenous manuscript. They start with definitions and sentences that were used (several times) before. Make the manuscript run smoothly, like only one person had written it, without multiple repetitions of phrases.

- Through the whole paper there is a lot of repeating of abrevations, like for example: inorganic polyphosphate (polyP),  Amyotrophic lateral sclerosis (ALS), frontotemporal dementia (FTD).... You declare the abbreviations only one time, the first time they are mentioned, that is the rule in scientific writing.

- Last but not least: less than 30% of the cited references were published in the last 5 years. This is much lower than expected for an up-to date review. Was there no new, fresh research published in the last couple of years? Please, check and add the relevant citations.

In my opinion, this Review presents a comprehensive report on inorganic polyphosphospate and the role it plays in cellular processes. But the manuscript could be notably improved by adding more explanatory figures and tables, by including the more recently published research papers, and addressing the other above given issues. After this is done, I would recommend its publication in the Special Issue "Polyphosphate (PolyP) in Health and Disease" of the journal Biomolecules.

Author Response

We would like to thank the referees for their constructive and useful comments. After going through additional analysis and re-drafting of the review, we feel confident that we have now clarified all their concerns. 

Please find below our point-by-point answers to all the comments. Our responses are in italics to make the text easier to read. Changes made in the new manuscript are highlighted in yellow.

We acknowledge all the observations. We believe that changes have strengthened the manuscript.

Respectfully,

Marcela Montilla and Felix A. Ruiz

Reviewer 1

R1: “The main problem with the paper in current form, in my opinion, is the lack of representative images and tables.”

A: In the revised version of the manuscript, we added 5 new figures.

R1: “…The abstract is too long and reads more like an Introduction than an Abstract. Make it shorter and focus on what is the highlight of this review, what has not yet been reviewed previously…”

A: The revised version of the manuscript, the abstract complies with the reviewers' requirement to shorten the text and focuses on the role of polyP in neurological diseases, which is the novel contribution of the review. The content has been reorganized according to its relevance to CNS pathophysiology.

R1: “Add a chemical scheme of polyP...”

A: The revised version includes a new Figure 1 with the chemical formula and spatial structure of polyP.

R1: “...Make the manuscript run smoothly, like only one person had written it, without multiple repetitions of phrases.”

A: The revised version of the manuscript has been completely reorganized, avoiding repetitions.

R1: “...there is a lot of repeating of abrevations”

A: In the revised version of the manuscript, we eliminated the repeated abbreviations.  

R1: “Was there no new, fresh research published in the last couple of years? Please, check and add the relevant citations”

A: In the revised manuscript, we have included a few new references from the last two years.   

Reviewer 2 Report

Comments and Suggestions for Authors

This manuscript presents an extensive, timely, and well-referenced overview of the emerging roles of inorganic polyphosphate (polyP) in mitochondrial bioenergetics, neuronal function, and neurodegenerative diseases. The authors have assembled an impressive body of literature, including both foundational studies and recent findings, and offer novel perspectives on polyP as a multifunctional molecule in neuronal physiology. The review is particularly strong in its depth of molecular detail and in highlighting the growing relevance of polyP in CNS-related pathology.

Nevertheless, several aspects of the manuscript could be improved to enhance its clarity, cohesion, and relevance to the stated theme. At times, the manuscript reads more like a collection of detailed data than a conceptually integrated narrative. While the scope is admirably broad, the central theme of polyP’s role in neuronal energy metabolism and neuroprotection is sometimes obscured by lengthy discussions of general polyP biology. A tighter focus on CNS-specific functions and mechanisms would improve the overall impact of the manuscript.

For example, Section 5, which discusses the regulation of mitochondrial function (pages 8–12), does an excellent job describing polyP’s roles in mitochondrial metabolism broadly, but it lacks contextualization within the nervous system. The word “neuron” is not mentioned at all in that section. Strengthening the link between polyP’s known effects on mitochondria and how these may manifest specifically in neurons would enhance the relevance of the review to its intended audience. Similarly, redundancy in content appears throughout the manuscript, particularly in discussions of mPTP regulation, AMPK/mTOR signaling, polyP’s chaperone-like activity, and its roles in ALS, FTD, and Alzheimer’s disease. Consolidating these recurring topics would streamline the text and improve readability.

In terms of scientific framing, the manuscript occasionally uses cautious or speculative language where stronger statements are justified. For instance, lines 56–59 describe how polyP has been “linked to” neurodegenerative diseases and “suggests” a role in calcium regulation. However, calcium buffering and mitochondrial calcium regulation by polyP are well-documented in several studies (e.g., Solesio et al., 2016; Abramov et al., 2007; Solesio et al., 2019). The use of “suggests” may inadvertently downplay established findings. Revising such language to reflect the strength of the evidence would improve scientific precision.

There is also an over-reliance on review articles among the initial references. For example, within the first 15 sources, a significant number (including citations 1, 2, 3, 4, 5, 7, 13, and 14) are reviews. While this is not inherently problematic, it makes it more difficult for readers to trace specific experimental data and understand the underlying methodology. Where possible, the authors should prioritize citing original research to support mechanistic claims.

Some important technical issues are mentioned but not fully developed. For instance, the manuscript briefly notes that “advances in super-resolution microscopy and molecular labeling strategies hold promise” for visualizing polyP (lines 330–332), but does not elaborate on current limitations of existing methods or why such advances are necessary. A few sentences clarifying what is currently lacking in polyP detection—and how that hinders further research—would strengthen the review’s forward-looking perspective. Similarly, on lines 895–897, the authors refer to experimental evidence that polyP reduces Aβ-induced cytotoxicity and alleviates paralysis in C. elegans models, but do not provide citations to support these claims. Unreferenced statements, especially when they describe experimental outcomes, should be avoided to maintain scientific rigor and transparency.

The visual elements of the manuscript also warrant improvement. While Figure 2, depicting polyP’s role in neurons, is conceptually helpful, it lacks sufficient mechanistic detail. Enhancing the figure to include more specific signaling pathways or molecular targets would make it more informative and valuable to readers.

In terms of structural suggestions, the abstract would benefit from being shortened and simplified, as it currently includes some redundancy—such as repeated mentions of therapeutic potential. The introduction should articulate the knowledge gap and central questions more explicitly and earlier in the section. Section 7, which discusses CNS disorders, overlaps heavily with Section 6 and could be more clearly structured by disease subtype. Section 9, which presents “highlights,” is largely redundant with the conclusion and might be better integrated into a unified final discussion focused on future directions.

In summary, the manuscript makes a compelling case for the biological and therapeutic significance of polyP in the nervous system, but would benefit from greater focus on neuronal mechanisms, clearer articulation of evidence strength, streamlined content, and improved visuals. The authors are encouraged to refine the narrative to keep the spotlight on neuronal energy metabolism and neurodegeneration, reduce redundancy in overlapping sections, cite more primary research, and clarify or expand underdeveloped points where needed. These revisions would help elevate the manuscript from a comprehensive summary to a conceptually cohesive and authoritative reference in the field.

Author Response

We would like to thank the referees for their constructive and useful comments. After going through additional analysis and re-drafting of the review, we feel confident that we have now clarified all their concerns. 

Please find below our point-by-point answers to all the comments. Our responses are in italics to make the text easier to read. Changes made in the new manuscript are highlighted in yellow.

We acknowledge all the observations. We believe that changes have strengthened the manuscript.

Respectfully,

Marcela Montilla and Felix A. Ruiz

Reviewer 2

R2: “...A tighter focus on CNS-specific functions and mechanisms would improve the overall impact of the manuscript.” /  “...redundancy in content appears throughout the manuscript…”

A: The revised version of the manuscript has been completely reorganized, reinforcing the idea of the importance of polyP in the CNS and eliminating repetitions. From the new section 4 “Multifunctional Nature of Polyphosphate in NEURONS”, until the end of the review, the focus has been reoriented on the functions of polyP in the CNS. Sections 4, 5, and 6 of the initial manuscript have been MERGED into this new section 4.

R2: “… the abstract would benefit from being shortened and simplified, as it currently includes some redundancy—such as repeated mentions of therapeutic potential…”

A: In the revised version of the manuscript, the new abstract complies with the reviewers' requirement to shorten the text and focuses on the role of polyP in neurological diseases, which is the novel contribution of the review. 

R2: “...The introduction should articulate the knowledge gap and central questions more explicitly and earlier in the section.”

A: In the new version of the manuscript, the introduction communicates earlier the knowledge gap and central questions.

R2:  “...Section 5, which discusses the regulation of mitochondrial function (pages 8–12), does an excellent job describing polyP’s roles in mitochondrial metabolism broadly, but it lacks contextualization within the nervous system. “

A: Sections 4, 5, and 6 of the initial manuscript have been MERGED into this new section 4, reorienting the revised manuscript on the functions of polyP in the CNS.

R2: “Section 7, which discusses CNS disorders, overlaps heavily with Section 6 and could be more clearly structured by disease subtype.”

A: Sections 6 and 7 of the initial manuscript were completely reorganized to avoid overlapping. In the new version, there are sections 4.8 and 5. 

R2: “Section 9, which presents “highlights,” is largely redundant with the conclusion and might be better integrated into a unified final discussion focused on future directions.”

A: Section 9 of the initial manuscript has been MERGED with the new conclusion on the revised version.

R2: “...lines 56–59 ...Revising such language to reflect the strength of the evidence “

A: Thanks for the observation. The paragraph was fixed in the revised manuscript.

R2: “...the authors refer to experimental evidence that polyP reduces Aβ-induced cytotoxicity and alleviates paralysis in C. elegans models, but do not provide citations to support these claims.”

A: Thanks for the comment. This reference was missing due to an unintentional error, which was corrected in the current version of the manuscript.

R2: “Where possible, the authors should prioritize citing original research to support mechanistic claims.”

A: In the revised version of the manuscript, we have changed some of the initial references. However, we have retained others since they are recent revisions, and we believe they can give a clearer idea of the general current state of knowledge in the area.

R2: “The visual elements of the manuscript also warrant improvement. “

A: In the revised version of the manuscript, we added 5 new figures, significantly improving it.